# Using ethics of care as the theoretical lens to understand lived experiences of caregivers of older adults experiencing functional difficulties

Kofi Awuviry-Newton[1,2]*, Meredith Tavener[1,2], Kylie Wales[1,3], Julie Byles[1,2,4]

1 Centre for Women's Health Research, The University of Newcastle, Australia, 2 African Health and Ageing Research Centre, New Castle, Australia, 3 Lecturer, School of Health Sciences, The University of Newcastle, Newcastle, Australia, 4 Global Innovation Chair in Responsive Transitions in Health and Ageing, School of Medicine and Public Health, University of Newcastle, Newcastle, Australia

* newscous@gmail.com

**Data Availability Statement:** Relevant data can be found in the manuscript. Data cannot be shared publicly because of participants' confidentiality and privacy consent per the ethics approval. Data are

## Abstract

The lived experiences of caregivers of older adults in Ghana are not well understood. The purpose of this study was to explore and discuss the lived experiences of these caregivers using the Ethics of Care as a theoretical lens and Interpretative phenomenological analysis as the methodological approach. Ten caregivers in receipt of social welfare services on behalf of older adults were recruited from the Social Welfare Unit at the Komfo Anokye Teaching Hospital (KATH) in southern Ghana. The analysis identified five interrelated themes: 1) committing the Self to caregiving; 2) caregiving impacting the Self; 3) motivating factors to caregiving; 4) caregiving burdens, and 5) thinking about personal affairs. Their experiences demonstrate that caregivers value the caregiving relationship, as posited by Ethics of Care, and tend to care for their health and well-being. Caregivers' expression of commitment to caring for older adults is mainly influenced by reciprocity, despite internal and external stressors, and desire to fulfil unmet personal needs. Ethics of care offers an understanding of the lived experiences of caregivers of older adults in Ghana. The findings draw attention to the state to develop specific programs to ensure the health, social and financial well-being of older adults' caregivers.

## Introduction

The population of adults aged 60 years or older in Ghana is growing, in both proportion and number, mainly due to decreasing birth rates and delayed mortality [1]. The number of older adults in Ghana increased more than seven-fold from 213,477 (4.5%) in 1960 to 1,643,381 (6.7%) in 2010 with the percentage further expected to increase to 9.8% by 2050 [2]. The growth in the number and proportion of the older adults population shows how Ghana is successful in healthcare, increasing the life expectancy for all Ghanaians including older adults.

Even though Ghanaians may be living longer, it does not mean they are free of aged related disability or frailty. Disability and frailty are significant influences on older adults' ability to

available from the University of Newcastle, Australia Institutional Data Access / Ethics Committee, contact via University of Newcastle Human and Ethics Committee, for researchers who meet the criteria for access to confidential data. Researchers may contact Ruth Gibbons (Human Ethics Officer), email address: Ruth. Gibbins@newcastle.edu.au, telephone: (02) 4921 6333, and URL: https://www.newcastle.edu.au/ research/support/services/human-research-ethics/ support-for-researchers/key-contacts.

**Funding:** The author(s) received no specific funding for this work.

**Competing interests:** The authors have declared that no competing interests exist.

function independently [1,3–5]. Biritwum and colleagues' study on disability (from the WHO Study on global AGEing and adult health (SAGE) project) established that about nine in ten older adults in Ghana experience functional difficulties [6] These functional difficulties include difficulty in engaging in daily activities for self-care, such as toileting [3,7,8] difficulty engaging in activities needed to live independently like preparing meals; and social participation activities such as attending social meetings and transportation [9–11]. Generally, older adults who experience difficulties in engaging in life activities often rely on their caregivers to fulfil their primary care needs. It is essential to explore caregivers' experiences of the assistance they provide to older adults who may be noticing a decrease in functional abilities.

## Background

Caregivers' importance is reflected in the promotion of their input in both health and social care since the 1970s worldwide [12]. Caregivers' relevance is incredibly profound in this era where the majority of older Ghanaians are expected to live with a disability in their later lives [13]. In the Ghanaian context, very often a caregiver is an adult child or a family member, who assists with all, or most of the older adults' needs [14–16]. The adult child caregiver is a relatively new phenomenon, resulting from a decline in the traditional extended family support system. The change in the extended family support system is primarily thought to have occurred due to migration, the modernisation of society, the introduction of formal education, population growth, economic hardship, and the arrival of numerous religious doctrines in Ghana [15,17–19]. To date, little is known about caregivers' experience of care provided to older adults or their coping strategies. Research on the lived experiences of caregivers will help in understanding caregiving sustainability.

Globally, substantial research exists regarding the factors that influence the care adult children provide to their ageing parents. An in-depth qualitative study in Sri Lanka revealed that adult children take pride in caring for their older parents [20], however, whether adult children provided the care out of personal interest or reciprocity was not explained. Earlier studies [21–23] identified psychological factors such as moral and religious obligations, attachment, and fulfilment of filial duty, societal expectation and affection toward their older adults as reasons for caring. Supporting more recent evidence, the current study identifies reciprocity, sense of obligation, selflessness, and feelings of closeness, secure attachment, and entitlement of care needs as some motivating factors to care for older parents. Moreover, being the only daughter, being the oldest child, and being closest to the older parent are also reasons for adult children care provision. Sons usually assume care for older parents when there is no adult daughter, or when adult daughters in the family live far away from the older parent or are experiencing their own chronic health conditions [24–27]. Irrespective of this evidence existing elsewhere, more information is needed in Ghana regarding the motivating factors influencing family caregivers' care for older adults.

Moreover, the literature on caregiving also reports on the decreased role of societal filial obligation, and a shift to personal moral judgement as a motivating factor to care and support older parents [28–30] For instance, a Japanese quantitative study found a high reduction in perceived filial obligation in daughters-in-law physical and emotional support, and a reduction in biological daughters' perceived filial duty in the provision of material support [29]. In Confucian Chinese society, where filial piety requires that daughters and daughters-in-law provide care and support out of respect and tolerance, caregivers report experiencing little or no motivation to assume this duty often due to a lack of proper parenting from their parent [27]. Holroyd's ethnographic study that focused on how caregiving daughters in China develop a sense of what is right and identified that daughters assume care work because of public reputation,

and moral obligation, more so than affection toward their frail older parents. For some adult children, they may provide support and care to their older parents for personal gains like obtaining inheritance [31].

In developed countries, positive effects of caregiving have been reported as an opportunity for caregivers to receive some advice and guidance from older adults, enhance social status, and increase their feeling of leaving a model for their children to learn from [32–34]. However, negative effects of caregiving have been reported, such as increasing caregivers' stress, inadequate well-being, poor psychological and physical health, and increasing financial difficulties [35–37]. In contrast, there are few studies on caregiving for older adults in Africa [38]. Although similar to the previous findings on caregiving, the available evidence suggests that caregivers of older adults in Africa do not plan to become caregivers but instead assume the responsibility and role change unprepared [39]. Caregivers have to give up work (such as farming), change for a different job, or miss work altogether to care for older adults [39–41]. Caregivers experience difficulty emanating from their low level of literacy, lack of paid work and the level of care needs of older adults [42,43].

In addition, the stress caregivers experience can lead to inadequate and/or poor quality care and support for older adults in Africa [9,39,41]. Many older adults report insufficient material support such as finances and food provided by children, irregular visitation by children and siblings, and in some cases, adult children taking too long in responding to the needs of the older adults [39,43–45]. Inadequate and poor-quality care and support have resulted from financial constraints on caregivers [42–44]. Despite the caregiving burdens, caregivers commit themselves to caregiving due to reciprocity and to negate any caregiving burden [46].

Presently, there is little qualitative evidence concerning caregivers' lived experiences of caring for older adults in Ghana, with available evidence-primarily concerned with caregiver income and financial constraints [44]. Quantitative evidence in Ghana revealed that caregivers experience emotional, health, and physical burden from caregiving [47,48]. Often less than 5% of caregivers receive financial, emotional, health, physical and personal care assistance [48]. A recent quantitative study conducted in Ghana revealed health and environmental-related factors that influence caregiver availability for older adults. These factors include: 1) older adults' advanced age, 2) being a widow, 3) living with a chronic condition, 4) hardly being understood by friends and family, 5) having no neighbourhood support, and 6) having two to four children [49]. The current study is unique because it seeks to capture, what it is like for caregivers to care for older adults who experience functional difficulties in Ghana, and to apply the Ethics of Care theory to understand their lived experiences.

## Ethics of care theoretical perspective

Ethics of care was used as a theoretical perspective to understand issues surrounding care provision for older adults in Ghana. Ethics of care recognises human relationships, interdependency and mutuality [50–52]. Rather than holding ethics as universal principles, ethics of care recognises that some responsibilities exist within certain relationships that do not exist in more general human interaction [52]. These duties and relationships are strongly gendered within most cultural contexts.

Most influential in the ethics of care approach are Gilligan [53] and Noddings [54]. Gilligan [53] studied men and women's attitudes and predisposition towards care as a construct of moral development, finding that men tend to adhere to ethical codes and principles, while women were more emotionally connected, and driven by interdependent relationships and concern for others. Noddings [53] extended this idea of the mutuality of caring relationships, believing that (for men and women) the ethical Self only exists in one-to-one caring

relationships and that the choice to enter a relationship hinges on the vision held by a person's best (ethical) Self. According to Noddings [54], the ethical ideal is intrinsic in the intimate relationship that two people establish for their own reasons. For Noddings [54], a person's best Self to care (ethical Self) depends on two factors. First is that a person has been cared for in past relationships; for instance, the care relationship between parents and children. Second, the person believes that they are in the best position to care for someone.

In the ethics of care, there are two roles: the cared-for and the caregiver. The caregiver enters with a receptive attitude without evaluation of expectations of the cared for and needs to be fully engrossed in the caring relationship [54]. They need to view care from the perspective of the cared-for, which deepens their understanding of the needs of the cared-for and strengthens the caring relationship. Noddings also talked about 'motivational displacement', where the caregivers ignore their own needs and concentrate more on how to help the one cared-for achieve what they need. Noddings also emphasised that receptivity (in any way) on the part of the one cared-for is essential to complete the caring relationship. According to Rice, acknowledgments does not have to be confirmation, appreciation, or reciprocal caring but has to be some manifestation of self-generated happiness that the caregiver can witness [55]. This acknowledgments will help caregivers to know that their efforts have been fruitful or valued by the cared-for.

Some researchers have criticised Noddings' theory of ethics of care, for several reasons. Hoagland [56], Houston [57], and Kyu et al. [58] have all argued that Noddings' ethics of care is dangerous because it lacks regulatory boundaries, and can lead to exploitation of caregivers especially those who are unpaid. Moreover, since many women are positioned in caregiving roles[53], the notion of caring as an inherently ethical part of certain strong (gendered) relationships creates gender inequality in care work. Keller [59], Koehn [60] and Meyers [61] criticised Nodding's theory for lack of autonomy of an assertion that a person's ideal Self is dependent on them fulfilling the caring role. This lack of autonomy reduces caregivers' ability to leave a distressing relationship when care is not reciprocated [59], making them more vulnerable. Due to this perceived lack of justice caregiving has the potential to be harmful and manipulative [60], denying caregivers the ultimate benefits of autonomy, self-respect and self-preservation [61].

Ethics of care served as a theoretical framework for this current study to help understand the motivations and vulnerabilities of caregivers in providing care and support for older adults in Ghana.

## 2. Methods

### 2.1 Design

This qualitative study is part of a larger concurrent mixed-method program of research exploring the functional abilities and care needs of older adults in Ghana. This current sub-study employed interpretative phenomenological analysis (IPA) and semi-structured interviews to gather participants' caregiving experiences. After analysing the interview data according to IPA principles, ethics of care was then employed as a lens to examine the findings.

According to IPA methodology, experiences are unique to the individual and require careful exploration and analysis to reveal nuances of the phenomena of interest [62]. Davidsen [63] recommends that critical reflexivity and engagement with interview transcript are needed to offer interpretations that reflect participants' experiences. Moreover, researchers need to declare their preconceptions and ideologies at the beginning of the work, and throughout the analysis stage for readers to be confident of the interpretations made [63].

In IPA, interpretations begin from the first interview with each transcript occurring consecutively. In this study, interpretations of findings followed three recommended stages. First, we offered interpretations of how participants understood their phenomenon. Second, we contrasted the interpretation of participants' experiences against the ethics of care. Third, in some cases, we raised general questions to make meaning of participants' experiences [63,64].

In this study, research question construction, sampling, data collection methods, interviewing and analysis followed IPA procedures [62]. Chan, Fung, and Chien [65] recommends four strategies to encourage bracketing. First, the literature was reviewed sufficiently to inform the research process. Second, the author responsible for data collection was open to learning about the experiences of participants. Third semi-structured interviews were used, allowing further probing during data collection. The fourth strategy, which usually involves the researcher returning to participants to confirm interpretations of their shared experiences, was not able to be achieved due to time and resource constraints. However, the primary researcher identified the factors, which could influence analysis and interpretation to reduce the biases in the study.

## 2.2 Sample and procedure

**2.2.1 Participant sample and site.** Purposive criterion sampling [62] was used to recruit caregivers seeking services on behalf of older adults from the Social Welfare Unit at the Komfo Anokye Teaching Hospital (KATH) in southern Ghana. Participants were eligible to take part in the study if they were older than 18 years, a caregiver providing care and support to an older adult (frail, sick, or incapacitated in any form) for at least six months, and provided informed consent. The caregiver did not necessarily need to be a child of the older adult receiving care. The study setting, Social Welfare Unit in KATH, was selected because caregivers seek and receive services from the Unit on behalf of those under their care, particularly older adults receiving health care at the hospital. The Social Welfare Unit is one of the many Units of KATH that seek to improve the psychosocial needs of patients and other health workers in the hospital setting. The Unit helps caregivers to make treatment decisions on behalf of older adults, counsels and offers emergency treatment for patients, including older adults whose relatives cannot be contacted immediately. The Social Welfare Unit only provides services to people attending KATH but does not receive a referral from the broader community.

For this study, caregivers were first identified by a receptionist at the Social Welfare Unit, using study eligibility criteria, as the caregiver presented seeking services on behalf of older adults under their care. A Research Assistant then shared the Participant Information Statement (PIS) and Consent Form (CF) with each caregiver and answered any questions they had about the study. The PIS and CF were explained in participants' language, Twi (the dominant Akan dialect in Ghana). Caregivers could provide consent to the Assistant Researcher at the Unit or to the primary researcher using a contact phone number provided on the PIS. Almost all eligible participants decided whether to participate within 72 hours. The primary researcher was informed of the consents and arranged an interview with each participant at a suitable place and time, within the hospital setting. Ten individuals provided written informed consent out of 53 eligible participants. According to IPA methodology, a sample size of 10 is sufficient to discover the nuances and complexities of people's lived experiences [66].

**2.2.2 Semi-structured interviews.** The primary researcher, who is experienced and trained in qualitative research, including IPA, interviewed each participant. Respect, concerns for privacy, a non-judgemental attitude together with genuine interest towards participants were always ensured. Participants were asked to reflect and talk about their experiences concerning the care they provide to older adults. Moreover, the primary researcher speaks the

same language as participants, (Twi), which helped facilitate interactions. The interview guide was developed specifically to encourage discussion by participants of how they experience providing care. Questions and prompts explored broad domains of the nature of caregiving, why they provide care, how they feel about their role as a caregiver as well as their coping strategies.

Each participant was interviewed once, with interviews lasting an average of 53 minutes. Interviews were transcribed immediately following completion. Transcripts were then coded and analysed. Codes were reviewed independently by co-authors, then discussed together as a group. Any discrepancies identified were resolved in turn through in-depth deliberations by all team members, with the consensus reached for each. During discussions with co-authors, definitions of identified themes were approved.

## 2.3 Quality of the research

The study was carried out in accordance with the four quality criteria formulated by Yardley [67], which are: 1) sensitivity to context, 2) commitment and rigour, 3) transparency and coherence, and 4) impact and importance.

First, to ensure sensitivity to context, we recruited participants according to the eligibility criteria; adopting an IPA approach was appropriate for the purpose of the study [62]; and the primary researcher was very informed regarding 'rules' surrounding social interactions and cultural beliefs and ideologies. However, this prior cultural understanding was bracketed by using Chan's four strategies Chan et al. [65] described earlier. Interview questions were uncomplicated and in lay language [68,69], also translated to Twi, the participants' language. Finally, the primary author showed respect to caregivers during interviews [68], listening to the information they wished to share without interruption. Respect to any person, in Ghanaian culture, takes precedence and is cherished [41].

Second, to ensure commitment and rigour, the primary author was attentive to the participant's information during data collection and careful in analysing each participants' transcript. Prior understanding of the IPA approach and research skills from previous qualitative studies increasing the author's commitment to conduct credible work. Interviews in Twi were transcribed and translated back into English; the transcription, translation, and analysis were all cross-checked by co-authors to ensure that participants' intended meaning was retained; and the primary author ensured disciplined attention to the inherent experiences in participants' interviews to understand how the participants made sense of their experiences. Also, existing literature was used to provide further context to the study [62].

The third criterion was transparency and coherence. For transparency, a thorough description of recruitment, analytical approach, and researcher's awareness of the relationship with participants have been provided in this study. To ensure coherence, authors were cognizant of the quality of narrative during their re-creation of the caregiver's lived experiences, in order for readers of the work to find meaning. Efforts were ensured to offer an interpretation that reflected participants' interview data.

The fourth and last principle is the impact and importance of the findings. Findings from this have demonstrated how healthcare and social care professionals' understanding of caregivers of older adults can be improved. Finally, the primary researcher undertook critical self-reflection to bracket his beliefs, and knowledge to reduce the impact it may have on the participant-researcher relationship.

## 2.4 Data analysis

First, the initial interview transcript was re-read several times. Second, after gaining an understanding of the first interview transcript, coding began. Third, the codes were then translated

into initial themes. In developing the initial themes, descriptive, in-vivo and process first cycle coding methods were employed [70]. Fourth, the initial themes were reflected upon to find connections between them through abstraction and subsumptions [62]. In terms of abstraction, like-themes were grouped together and named. With subsumptions, some themes were put under others because they fit well. These processes allowed for re-organisation, categorisation and further analysis to further enrich the analysis (Morse, 1994). A table specifying final themes and respective sub-themes was then developed for the first transcript, reviewed and discussed. Fifth, the previous steps one through four were repeated for each one of the remaining nine interview transcripts. The sixth step was to look for patterns of themes across all ten transcripts, leading to a master table of five overarching themes (see Table 2 in Findings). Co-authors compared the five overarching themes to the individual transcripts to ensure they reflected the participant's statements. The five themes were then reported in the form of a narrative account supported by participant quotes.

## Ethical consideration

Ethical approval for this study was obtained from The University of Newcastle (Australia) Human Research Ethics Committee (H-2018-0148), and Kwame Nkrumah University of Science and Technology (CHRPE/AP/112/18) in keeping with the Declaration of Helsinki. Informed consent was obtained from the study participants. Anonymity and confidentiality were ensured.

## Findings

Participants ages ranged from 25 to 60 years, and all but one identified as female. Participants were providing care to their own family member mostly parent for at least seven months. Caregivers provided the care to older adults with varied health conditions (see Table 1).

The five interrelated themes were: *1) committing the Self to caregiving; 2) caregiving impacting the Self; 3) motivating factors to caregiving; 4) caregiving burdens*, and *5) thinking about personal affairs in addition to caregiving*. These themes together reflect the lived experiences of caregivers (Table 2).

### Committing the Self to caregiving

This theme refers to the inherent acts, duties and behaviours that caregivers ensured while caring for their older adults. Caregivers' dedication was reflected in the extent to which they

**Table 1. Demographic information of family caregivers.**

| Pseudonym* | Gender | Age | Occupation | Caregiving duration | Care recipient, age | Care recipient, condition(s) |
|---|---|---|---|---|---|---|
| Margaret | Female | 47 | Bread seller | 10 years | Own father, 71 | Stroke |
| Cecilia | Female | 58 | Farmer | 7 months | Own mother, 100 | Bedridden resulting from advanced age, h Hypertension, diabetes |
| Naomi | Female | 31 | Popcorn business | 5 years | Own mother, 70 | Diabetes, hypertension |
| Gloria | Female | 60 | Unemployed | 7 years | Grandmother, 83 | Hypertension and advanced age |
| Cynthia | Female | 23 | Unemployed | 2 years | Grandmother, 88 | Diabetes |
| Eugenia | Female | 34 | Cosmetic seller | 5 years | Own mother, 65 | Incontinence (urine and bowel), |
| Davida | Female | 34 | Shop owner | 10 years | Cousin, 72 | Incontinence, diabetes |
| Phinehas | Male | 25 | Driver | 5 years | Grandmother, 76 | Incontinence, stroke |
| Cindy | Female | 57 | Operate provision shop | 15 years | Own mother, 94 | Hypertension, stroke and diabetes |
| Elizabeth | Female | 42 | Farmer | 7 months | Own mother, 73 | Incontinence and stroke |

*names used in this study are false names

**Table 2. Superordinate themes and sub-themes reflecting caregiver experiences.**

| Superordinate Themes | Sub-themes |
|---|---|
| **Committing the Self to caregiving (#1)** | Caregiving duties |
| | Extent of caregiving |
| | Feeling about the continuity of care |
| | Managing caregiving stress |
| | Sacrificing Self for cared-for's needs |
| **Caregiving impacting the Self (#2)** | Caregiving as a cost to caregiver's Self |
| | Caregiving as an opportunity to benefit the Self |
| | Priding Self over caregiving |
| **Motivating factors to caregiving (#3)** | Caregiving as obeying God |
| | Caregiving as setting an example for children |
| | Caregiving as a duty |
| | Fear of social censure |
| | Caregiving as reciprocating the good |
| | Caregiving due to a permissible condition |
| **Caregiving burdens (#4)** | External stressors |
| | Stressors surrounding caregiving |
| **Thinking about personal affairs (#5)** | Concerned about improving the future for Self |
| | Need to strengthen the relationship |
| | Pushing for a caregiving relief |

provided care, the duties they performed, the self-sacrificing spirit towards older adults' needs, the desire to manage caregiving stress, and their strong wish to continue caring. Some caregivers negotiate the care they provide with their siblings, while others provided the care alone. Central to Cecilia's statement was the desire to share caregiving duties:

We decided that every month, someone would be there and care for her. This month, for instance, I am the one who is taking care of her. When I am done someone else will come and take over.

For those caregivers who negotiate care with their siblings, it appears that taking on all caregiving duties by themselves when it was their turn, was an expression of total commitment to the fulfilment of the needs of older adults. See how Cecilia emphasised each sibling's awareness of how they prepare to care:

As for that, when it reaches an individual's turn, they prepare to care for the older woman alone. Therefore, when it gets to my turn, too, I solely care for her.

When there is no such arrangement about how, who and when to care, the decision then depends on the closeness and prior relationship between a family relative and older adult. It is on this relationship that decision to care hinges, even if prior negotiations for caring existed. Gloria explained how she decided to care for her grandmother, intuitively:

She was doing a chop bar work cooking food for people to buy. That time she became weak, and so she was not able to work, so I chose to take over from her and run the chop bar business. Also, because all her children had travelled, I decided to stay with her and take care of her.

Another topic of discussion was caregiving duration. Commitment, to caregivers, meant spending many hours and years caring for older adults. Margaret talked about how long she had been providing care for her father's needs:

I have cared for him for more than even 10 years. And at his present condition (stroke), I am still caring for him.

For Cynthia, she dedicated many hours to her mother's care:

When I wake up in the morning, at 6 am, I will wake her up to take her diabetic drug and at 6:30 am, she can eat her food. Then at 10 am, I will have to give her fruit. And at 12 pm, I will have to give her good food. At 2 pm she will eat fruit, at 4 pm I give her good food, and this continues throughout the night.

Commitment to caregiving was evident in the number of older adults' who were receiving care, from one caregiver, whilst not the primary care recipient. To commit oneself to caregiving seemed to entail being ready to offer support to anybody who needed it. From the extract below, Elizabeth described her care for others aside from the primary care recipient:

I provide care to other people. Even now, I have my mother and her sister at the hospital I care for them.

At the time of her interview, Naomi was taking care of two older adults, including her mother, who suffered from multiple chronic conditions:

I care for two people. They are two siblings I am caring for them in the same room: my mother and her older sister.

Commitment to caregiving meant accepting care duties irrespective of nature. Cecilia elaborated on how she assisted her mother in self-care:

When I wake up in the morning when I finish bathing her, she being an older woman, she becomes hungry quickly because of the numerous drugs she has been taken. So, I will have to cook and give her some to eat and keep some there for her so that anytime she request for food, it will be ready at the right time.

Eugenia explained what she does to assist her mother with incontinence:

She goes to the toilet on herself. Because of this, I make her wear diapers. Her condition has worsened, she cannot even tell me that she will want to urinate or defecate and so she can defecate or urinate on herself. With that, I will have to wash all the bedsheet using Dettol so that there will not be any scent on her.

Phinehas, the one male caregiver in this study, spoke about how he assisted his grandmother in bathing and dressing:

If she wants to bath too, we have a chair in the bathroom, and so I will assist her to sit on it and bath. When she is done bathing herself, I will pour water on her, and when I am done, I will clean the water on her. Then I will clothe her and bring her inside the room.

Commitment to older adults' included assistance with medication, personal care, household chores and general conversations. To commit to caregiving also meant ensuring that the needs of older adults came first, and caregivers' personal needs, came second. Caregivers demonstrated appreciation for human life and relationships. It is in this context that commitment to caregiving comes in to play expressively. Cecilia chooses 'life' over 'work' as an expression of her commitment to caring for her older mother:

Human life is very important, as for work, it is there at any time.

Appreciation for the dignity of human life forced caregivers to work fewer hours. Naomi described how she reduced her time working as a popcorn seller in order to care for her mother and her sister:

Both of them have diabetes, and both of them have a time they use to eat, and if I don't prepare food for them on time they will not eat, and if they don't eat early, it can make their condition worse at night and so I had to close from work and come early to care for them.

Commitment to caregiving was reflected in the caregivers' management strategies to combat their own stress to ensure continuity of care. To some caregivers, commitment meant being spiritual. Caregivers put their faith and hope in God as a medium of lessening the effects of caregiving burdens. Resorting to prayers, Cindy expressed renewal of strength to provide care:

Sometimes, I will feel about quitting the care, but when I think about God and pray, it helps me to keep taking care of her because God gives me encouragement.

It was revealed that hope for good things empowers caregivers to remain in caregiving while they put personal desires or demands on hold. Margaret expressed worry when she reflects on the neglect of her own family to provide care to her father, but then she feels consoled with her trust in God for blessings:

When it gets to some point, it worries me most that I have left my family, my children and husband there to assume this duty. When it gets to certain times, I comfort myself with the hope that God will bless me. Whatever God does is good.

For Margaret, commitment to caregiving meant maintaining her focus on caregiving duties:

What makes it easy is that when I get close to him, I concentrate on all what I suppose to do for him, and this make it easy for me.

Social support offered a boost when providing care to older adults. It appears that government support was only dependent on older adults' contribution to social security (i.e. a pension). None of the caregivers reported receiving support from the government. Support from church members increased caregiver's willingness to continue. Phinehas reflected on how church members' encouragement helps him to cope with caregiving stress:

You see in my church, I have one father who likes me much, and so he sometimes comes to the house, advises me and encourages me to go on. It is because of his encouragement and advice that I am still here today.

Support from family and friends appeared to be the main available support for caregivers in meeting the needs of older adults. Margaret illustrated how she receives support from her husband:

> Concerning my husband, he understands that I am caring for my father here. That is why he allowed me to come here and care for my father.

Caregivers expressed willingness to continue providing care, even though feeling burdened in their role. For some caregivers, caring for an older adult was a lifetime duty. Reflecting on the extent to which she desired to provide care, Gloria explained:

> It is only death that can part us. It is God who gives and takes, so if the time reaches and God takes her, what can I say? Other than this, I will continue to take care of her.

## Caregiving impacting the Self

For caregivers, caregiving was a period of self-transformation. At one point, caregiving was a blessing; at other times, it was a cost. Perception of caregiving, in the minds of caregivers, takes the form of a chameleon, where its true form becomes challenging to pinpoint and describe. Caregiving, at any point in time, was a combination of mixed feelings.

For all caregivers, providing care had at some time, come at a personal cost. It seemed that care duties and caregiver's poor health are inseparable. Cindy's account below is an illustration of how caregiving to her older mother has impaired her health:

> I feel pains. When she was admitted to the hospital, I have suffered a lot like when I sit in a car from a long place morning and night. I am risking my life.

It was always a choice between caregiving and precious time with friends. The desire to maintain friendship always manifested itself as an issue of concern for caregivers. It is in this context that Cindy became anxious about the possible impressions, her friends hold regarding her limited time with them:

> Oh, sometimes I think that my friends may think that maybe someone has said something bad about them to me, and that may be the reason why I am not visiting them again.

Caring for older adults was incompatible with maintaining caregivers relationships with immediate family. It was in this context that caregivers felt irresponsible for failing to undertake their role in the family. For Cecilia, caring for an older relative implied neglecting some crucial roles as a mother who was expected to provide food for her immediate family:

> When I was living with the family, I use to cook for them to eat and now that I am not there, they sometimes miss me.

For Margaret, taking care of her father away from home affected her children's education, spiritual growth and feeding:

> I have children at home, and for now, even regarding my children schooling, it has become something bad. As for a mother, if you are at home, you will know how to cater and nurture your children, but because of my father's illness, I have moved from home and come to stay

in Kumasi. For my children welfare, I cannot say now. However, this is not better than when I was there with them. Even during Sundays, maybe my children don't even go to church, but if I were there, it would not happen this way.

For Cynthia, caregiving has dissociated her from providing support to her own parent:

I am not even able to go to my family, which is my parents, to visit them.

Caregivers described how caregiving disrupted their work causing financial constraints. Sustaining work and finances while providing care seemed impossible. Inherent in Phinehas' account was a demonstration of the devastating impact of caregiving on finances and work:

When she became weak, and I started taking care of her, I have lagged behind so many things. At first, I was able to earn about Gh2000 (US$400) every month, but for now, because I left the work for other people when the money comes, I share the money with those people.

The second dimension under the transformation of self was the perceived benefit irrespective of the inherent cost to the self. Gloria illustrated how she benefited from caregiving financially:

When her children come, they sometimes give me money.

To some caregivers, caregiving was a medium for exposure to new skills and knowledge in life. In the following extract, Cecilia illustrated how she benefited from caregiving, **emphasising** how her mother was seen as a reservoir of knowledge and full of advice for life:

This is because any time I come home, I see her, and she advises me, and we converse all the time, and that makes me happy. Whom will I go to if she was dead?

Caregivers felt happy when they realised that their caregiving duty was fulfilled, working from the assumption that care improves the older person's life, even given the cost of caregiving. For Margaret, it appeared that her happiness depended on their older relative being alive:

This is what I am saying that people suffer, yet they lose their older adults they care for. But in my case, with all the suffering, he is still alive. This has made me know that I have gotten some benefit.

Caregiving seemed to offer spiritual benefit for Davida:

Last time one pastor said the care I provide for this woman is protecting me. He said some people want to kill me, but because of the care, I provide for this old cousin God has been rescuing me.

## Motivating factors to caregiving

This theme encompasses the aspect of caregivers' experiences that motivated them to remain as carers in the face of caregiving challenges. It appeared that reciprocity drives other motivational factors to operations. Caregivers demonstrated varying but related reasons for providing care. Reciprocity was present at every expression of caregivers' even up to the point where

expressing their desire to quit caregiving. Elizabeth described how she could not quit continuing to provide care:

> Because she was washing my clothes and cleaning my toilet when I was young, and she was taking care of me so if she did not use gloves, then I will not use gloves in cleaning her toilet.

Caregivers perceived caregiving to an older adult as a duty that needed fulfilment by any possible means. Fundamental to this belief was reciprocity. Cindy emphasised how her reason to care was influenced by a sense of duty to care:

> We live in the same house with my mother and because my mother brought me to this world and so, it is my duty to continue caring for her.

The desire to obey God was at the forefront of some caregivers' narratives. To caregivers, God takes an interest in their adherence to caregiving duties for older adults. It appears that the desire to obey God could not be separated from reciprocity. Margaret spoke of this link, expressed as a fear of God and reciprocity:

> I will have to obey God because the bible says children should obey their parent in the Lord. And so, I don't worry using my time and my money to care for my mother because If I don't do that, I will get punishment from God.

The recognition that one day, caregivers may, in turn, require care from their children held great meaning according to caregivers' narratives. Margaret explained her belief in reciprocity and how that influenced her desire to leave a model for her children:

> When it gets to some point, I think that when parents enter a distressing situation, I have to devote myself to care for him. If you don't care for him one day when you grow old, your child will do the same thing to you.

Caregivers sometimes provided care because of circumstances such as being the only daughter among siblings, being the eldest daughter, and self-employed. Eugenia talked about how being the eldest daughter obliged her to accept the caregiving to her older mother:

> Out of all my mother's children, I am the elderly daughter among the daughters. Although I don't do any rigorous work or work that requires much time, I am a cosmetic seller. Because she is my mother, I needed to forget about my work and keep taking care of her because she is sick.

Another motivating reason was caregivers' concerns about other people's approval and the need to keep their relations with significant others strong. This fear was like a glue that bonded them to their caregiving duties. Central to a fear of social censure was the ever-present expression of reciprocity. Anticipating the likely fear of ridicule when dissociated self from caregiving, was Naomi's reason to care:

> What motivated me is that she is my mother, I don't have any other mother elsewhere, and if I decide not to care for her, people will even say bad things about me.

## Caregiving burdens

This theme reflects the different external and internal stressors that can burden caregivers. There appears to be a double blow, one from the caregiving duties and another from external stressors. External stressors are the felt tension that emanates from older adults, unfriendly environments, and caregivers' siblings. These stressors place a burden on caregivers. Margaret described the frustration she experiences emanating from her older father's uncooperative attitudes:

> I have two Kuraba (chamber pots) I have given to my father, one that he needs to urinate in and the other he needs to spit in but it will get to a time where the lid will be on a chamber pot, but he will spit on the lid. It will get to a time when you will see that he wants to pass faeces, I will ask him several times, but he will tell me that he will not go. When hospital official tells us to leave the wards, it is there that he will call me and tells me that he wants to go to the toilet. Therefore, when it gets to this time, it makes the caregiving very difficult for me.

The physical environment appeared to serve as a constraint for caregivers. For Cindy, who has been caring for her older mother for nine months at the hospital, the unfriendly nature of the hospital stairs cause her problems?

> Coming to this hospital and climbing stairs up and down make it difficult for me too.

Caregivers demonstrated feelings of tension concerning the pressures from their siblings or family members. Davida expressed worry about the false rumour her family members spread about her concerning the care she provides for her older cousin:

> Sometimes her sisters can say that I have intentionally decided not to allow anybody to come and stay in the house and so I have decided to take care of her alone for some benefit, and that worries me.

Naomi emphasised the tension she feels from her husband alluding to her unfulfilled duty as a wife as causing an impediment to care:

> First, has to do with the tension between my husband and me. There is no peace between us at all. The reason is that, when my husband needed me most, I will not get time for him because I am caring for my mother. Therefore, all the time, he will be thinking about me wrongly, he will not even talk to me dearly, and it is always quarrelling. Sometimes, he also insults me when I call him on the phone.

Internal stressors included the enormity of the care duties, making caregivers feel burdened. It was evident in their comments that assisting with toileting and vomiting overburdened them. This theme also encompasses how intimate caregiving was causing an unpleasant feeling for caregivers. Reflecting on the past and the older adults' current health problems, Cecilia admitted an increase in caregiving responsibility:

> When she was not sick, she used to help me out in certain things like picking chairs and bowls. She is growing older, and because of the illness, she cannot even help me out in anything. Therefore, I have to do everything for her. I change her clothes every day.

Although caregivers expressed the willingness to care, they found assisting older adults with toileting and cleaning to be the most unpleasant. Elizabeth shared the difficulty she encounters when she compared assisting toileting in young children and that of older adults like her mother:

> You know that as for here this town, we are used to giving children a chamber pot. For me it is much easier to clean toilet of children than that of adults. Therefore, it was difficult for me to clean the toilet of my mother.

Davida spoke of assisting with incontinence and other daily living assistance with unhappiness and unease:

> It is the toilet and urinating herself that makes it difficult for me. Even as for feeding her, it is not a problem for me, but the toilet is a problem for me. Every day, she defecates on herself, and I will have to use my hands to clean the toilet.

Eugenia was different from others because, for her, assistance with vomiting overburdens her more than with toileting:

> As for me, it is the vomiting and the toilet that becomes difficult for me to do. Out of these two, the vomiting is more challenging to perform.

### Thinking about personal affairs

Despite caregivers' desire to surrender themselves to the care needs of older adults, they concurrently described behaviours and attitudes connoting internalised concerns for their welfare. Expressions of the desire to want to improve their future became a matter of concern primarily when they reflected on their unfulfilled dreams. Referent to God's approval for her to seek personal blessings and the need to further her education, Cynthia talked about her future:

> A time will come that I will stop. Because I also have to look for my future in terms of education. Although it is good that I care for her because I can get God's blessings, yet I have to look for my future. Spiritually, God has blessed you, but I will have to try to make sure I take a step for what God has purposed about me to be fulfilled.

For Davida, she wished for a husband to live with. She reflected on how a potential husband might interpret caregiving to her older cousin:

> Even if the man wants us to go out after marriage, I will not be able to leave my older cousin behind. Unless I get a man who will understand that, he will come and live with us in the same house with my older cousin. Even with that, a man can say that because of the toilet and other things I clean, he will not even eat the food I prepare.

The desire to strengthen relationships with significant others was discussed by caregivers. Though Margaret accepted caregiving to her father, she expressed concern and the need to improve her relationship with her immediate family:

> This is because I have little children and a husband. Therefore, leaving all these people and coming to stay and cater to my father, it was not easy at all. However, because there was nobody available, I decided to care for my father.

Caregiving pressures, together with personal reflections on their unfulfilled aspirations, usually led to a push to seek a break from caregiving. It is in the context of a push for a break that Cynthia seems to achieve a result:

Always I tell them. Even my father said next year February when all his brothers come from abroad, and he said he would have a meeting with them that next year I will have to go to school.

Naomi expressed the wish to share the caregiving duty with a paid caregiver:

If I get financial help, I can even hire one person to assist me in caring for them. In that case, I can go outside and work small while the person cares for them.

## Discussion

This study employed the ethics of care theoretical framework to understand the lived experiences of caregivers of older adults who required help with everyday activities in Ghana. In this study, caregivers express commitment to caring for older adults motivated primarily by reciprocity, despite internal and external stressors, and the desire to fulfil unmet personal needs of caregivers. Ethics of care offers an explanation to experiences of caregiving to older adults in Ghana [50]. Applying the ethics of care in a developing country like Ghana, we were able to identify that caregivers are also conscious of meeting their demands despite caregiving commitment determined by relationships.

Caregivers' appreciation of the relationship between themselves and older adults was demonstrated in this study, reflecting the ethics of care framework as valuing mutuality, interdependency and people coming together in a caring relationship [50–52]. The decision about sharing caregiving duties among siblings, spending many years providing care and putting aside their own needs and wishes connoted the extent of commitment they gave to caregiving to older adults. Similar findings were revealed in Nigeria, which has a similar cultural context in terms of relationship and mutuality [46]. In Faronbi et al [46] study, caregivers manage caregiving challenges and assist older adults with chronic conditions as an expression of caregiving commitment. This current study adds that caregivers show a willingness to continue caring for older adults until older adult receiving the care dies signalling the extent of their commitment level. These findings are similar to the evidence from Sri Lanka, a developing country with similar socio-economic context as Ghana, those adult children take pride in caring for their older adults [20]. The finding from this study show how caregivers in Ghana, despite their demands, are committed to meeting the needs of their older adults. Sometimes the over-commitment of caregivers in meeting the needs of older adults can lead to caregivers not satisfying their own needs and demands. Given this suggestion, the major concern for policy direction in Ghana should be how specific intervention can be developed and sustain the long-term care older adults receive. We assert that without the state' support for caregivers in Ghana, irrespective of caregivers willingness, they may not be able to continue to provide for their older adults. Specific programs including provision of financial support, are relevant to ensure continuity of care.

The finding from the present study shows that caregiving can have both negative and positive impacts on caregivers. Caregivers openly acknowledge the cost caregiving had on all aspect of their lives, which is in contrary to other finding that caregivers can deny burden from caregiving [46]. The cost to caregivers manifested itself in poor health, impairment of friendships, self-blaming, perceived loss of role as a parent, couple, or child, negative impact on work and

finances, feeling overwhelmed that life visions will remain unfulfilled. The adverse effects found in this study are similar to the effects described for carers in developed countries [35–37]. These similar findings suggest that the negative effects of caregiving on caregivers may not be contextual; however, the management of these effects may be impacted by the level of the country's wealth and support structures. These findings will be relevant for stakeholders, including social workers, to improve the health and social needs of caregivers. Additional support for caregivers to mitigate caregiving stress, including counselling and respite services, may assist in balancing care responsibilities with the caregivers' own life events, including working, attending school and fulfilling their marital duties.

Addition to the above factors burdening caregivers found in this study, care to the opposite gender, the intensity of the care duties, unfriendly physical infrastructures, uncooperative attitudes of older adults and influence of family members intensify the internalised feelings of burden. Caregivers need to be trained on older adults' needs and how the care they provide can enhance older adults' functional abilities. With appropriate training, caregivers can provide care to minimise caregiving burdens. Given the economic condition of most of the caregivers in Ghana, it will be impossible for caregivers to afford the cost of the training. Moreover, caregivers may not be aware of the need to receive training on how to care. These challenges can be met if the state assumes responsibility to encourage caregivers to enrol in caregiver training programs at subsidised prices. The training can cover care needs of older adults, how to engage older adults in their care and the need to be conscious of the physical and social environment impacting on older adults health [71].

One factor increasing the caregiving burden found in this present study was assistance with toileting. This finding is consistent with studies showing that having toileting difficulty increases older adults dependency on caregivers for support and care [3,72]. On the other hand, the perceived benefits from caregiving were an opportunity for carers to enhance and enrich themselves by developing new skills and knowledge, receiving spiritual protections, receiving social approval, and raising self-pride. Similar benefits have been described in other studies [32–34]. The perceived benefits found in this current study offer caregivers some reason to hold on to caregiving despite the inherent cost. Although inherent cost exists in caregiving, caregivers feel self-fulfilling completing caregiving roles [50]. This perception may lead to lack of autonomy, and reduces caregivers', mostly women, ability to select and leave a distressing relationship making them more vulnerable [59–61]. Supporting caregivers and allowing them to fulfil their life endeavours can promote their well-being, ensuring the continuity of care in Ghana.

The study shows that several factors motivate caregivers to care for older adults. Inherent in these motivations was the sense of reciprocity as depicted by ethics of care [50]. Caregivers provided care to older adults motivated by seeing care as a sense of duty, obeying God, leaving a model for their children, less demanding circumstances, and concern about people's approval all driven by a sense of reciprocity. The findings in this study corroborate with other studies revealing reciprocity as the primary motivation influencing adult children to provide for their older adults [24,27,41,44,46]. It appears that in the Ghanaian context, caregivers' sense of accomplishment, fulfilling care duties depend on the awareness that they were cared for in the past relationship. Though the current study has highlighted adult children's willingness to continue providing care, their motivation may not be continuously be hinged on reciprocity or sense of duty as urbanisation seems to be rapid in Ghana. Adults children may resort to providing care based on personal judgement or choice, or personal gain just as reported in China and Japan [29,31]. A decline in perceived obligation to care for their ageing parent is expected in Ghana given the enormous burden on caregivers, including their inability to fulfil their own needs and demands. A reflection in this perspective is a result of some older

adults reporting that they are being deprived of care and protection in Ghana [9]. The current study's finding implies that the government should assume authority over the long-term care for older adults, requiring that caregiver's health and social needs be ensured to ensure continuity of their care to older adults. Providing caregivers with training, health, financial and social care can foster their reciprocity motive, which may increase their willingness to remain in care for older adults.

Caregivers demonstrated an awareness of the need to fulfil or accomplish their own personal needs. Personal needs involved continuous education, getting married, strengthening the relationship with significant others such as husbands, friends and family, and pushing for some respite from caregiving. Though caregivers concentrate more on how to help older adults achieve their needs [50], this current study adds to the ethics of care that caregivers do not totally ignore their own personal needs but rather are conscious of fulfilling them. The unfulfilled vision will create more anxiety for caregivers, which will have a negative impact on caregiving. Given the long-term care nature in Ghana, where family especially adult children provide care with no or minimal government support, it will be very challenging and more stressful for caregivers to provide long-term care for older adults in Ghana. These findings call for the government of Ghana to implement the policies in the national ageing policy [73], that seek to ensure the health, financial and social needs of caregivers. Currently, the social protection programs, which includes Livelihood Empowerment Against Poverty in Ghana do not adequately meet the needs of eligible older adults, let alone factoring the needs of caregivers of older adults. Social workers can be very helpful in developing programs that can help caregivers achieve their needs [74]. In this regards, a respite care systems can be instituted to support caregivers in caring for their ageing parents.

Our study findings have numerous implications. First, nationally instituted caregiver benefits should be established to support caregivers of older adults in Ghana to serve as a demonstration of how care provision is valued [55]. Second, per the difficulties caregivers experience with infrastructures, user-friendly public transportation, facilities, and accessible health care services for caregivers need to be established and implemented to support people providing care. Third, the study indicates the importance of promoting spirituality among caregivers to improve their mental and emotional well-being. Fourth, sibling relationship with primary caregivers on care for older adults warrants further investigation, with the potential to enhance emotional, instrumental and financial support provided to the caregiver. Moreover, more research with male caregivers is needed to understand how caregiver experiences differ among men and women.

The strengths of this study include the application of ethics of care into exploring caregivers care to older adults. Second, the study recruited caregivers who provided care to older adults for at least six months serving as a common ground for providing in-depth experiences of caregiving. One limitation was that having only one male caregiver was not enough to understand how caregiving roles playout for men who provide care.

## Conclusion

Caregivers demonstrate a commitment to caregiving despite several stressors and costs to themselves. The findings reveal how caregivers' care for older adults are being fostered by reciprocity. Irrespective of this, the caregiver is concerned about fulfilling their life endeavours and needs. This study recommends that the state assuming authority in promoting social support, respite care system, financing long-term care, improving transportation, healthcare and user-friendly public infrastructures if put in place for caregivers, could help to replenish the effort caregivers expend.

## Supporting information

**S1 File. Interview schedule.**
(DOCX)

## Author Contributions

**Conceptualization:** Kofi Awuviry-Newton, Meredith Tavener.

**Formal analysis:** Kofi Awuviry-Newton.

**Investigation:** Kofi Awuviry-Newton.

**Methodology:** Kofi Awuviry-Newton.

**Project administration:** Kofi Awuviry-Newton.

**Software:** Kofi Awuviry-Newton.

**Supervision:** Meredith Tavener, Kylie Wales, Julie Byles.

**Validation:** Kofi Awuviry-Newton.

**Visualization:** Kofi Awuviry-Newton.

**Writing – original draft:** Kofi Awuviry-Newton.

**Writing – review & editing:** Kofi Awuviry-Newton, Meredith Tavener, Kylie Wales, Julie Byles.

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
