## [Decision Letter · Decision Letter 0]

10 Jan 2022

PONE-D-21-02211

Using Ethics of Care as the theoretical lens to understand lived experiences of caregivers of older adults experiencing functional difficulties.

PLOS ONE

Dear Dr. Awuviry-Newton,

Thank you for submitting your manuscript to PLOS ONE. After careful consideration, we feel that it has merit but does not fully meet PLOS ONE’s publication criteria as it currently stands. Therefore, we invite you to submit a revised version of the manuscript that addresses the points raised during the review process.

The manuscript has been evaluated by two reviewers, and their comments are available below.

The reviewers have raised a number of concerns that need attention. They request additional information about the conditions experienced by the care recipients, a more in-depth interpretation of the interview extracts and clarification of some statements.

Could you please revise the manuscript to carefully address the concerns raised?

We look forward to receiving your revised manuscript.

Kind regards,

Lorena Verduci

Staff Editor

PLOS ONE

Journal Requirements:

3. Please include additional information regarding the interview guide used in the study and ensure that you have provided sufficient details that others could replicate the analyses. For instance, if you developed a questionnaire as part of this study and it is not under a copyright more restrictive than CC-BY, please include a copy, in both the original language and English, as Supporting Information.

5. Your abstract cannot contain citations. Please only include citations in the body text of the manuscript, and ensure that they remain in ascending numerical order on first mention.

Reviewers' comments:

Reviewer's Responses to Questions

**Comments to the Author**

1. Is the manuscript technically sound, and do the data support the conclusions?

Reviewer #1: Yes

Reviewer #2: Yes

2. Has the statistical analysis been performed appropriately and rigorously? 

Reviewer #1: N/A

Reviewer #2: N/A

3. Have the authors made all data underlying the findings in their manuscript fully available?

Reviewer #1: No

Reviewer #2: No

4. Is the manuscript presented in an intelligible fashion and written in standard English?

Reviewer #1: Yes

Reviewer #2: Yes

5. Review Comments to the Author

Reviewer #1: This manuscript describes the findings of a phenomenological study of the experiences of caregivers of older adults in Ghana. The authors present a well-written summary of their findings with in the context of the Ethics of Care.

Additional information about the conditions experienced by the care recipients would help to put the caregiving narratives into context given that caregivers of individuals with different chronic conditions experience caregiving and its strain differently. For example, caregivers of people with dementia often rate their strain/burden and stressors higher than those caring for adults with other conditions.

Reviewer #2: This manuscript presents a qualitative study that explored the lived experiences of caregivers of older adults with functional difficulties. The findings are insightful and the manuscript is also well written. My main concern with the manuscript is the presentation of findings. The findings generally depict detailed experiences of caregivers; however, given the use of the IPA analytical approach, I thought the authors could enrich the presentation by providing a more in-depth interpretation of the interview extracts used to support the narratives. Rather than leaving extracts hanging, it would be helpful if efforts were made in interpreting the subjective meanings of the extracts to the participants. Below are a few (but not the only) instances I thought more could be done with the extracts:

1. On page 18 (last line) it is indicated that “Commitment to caregiving was reflected in the caregivers’ management strategies to combat their own stress …” However, the specific strategies the caregivers employed in managing their stress are not indicated nor elaborated. Moreover, I was wondering why these stress management strategies could not be captured as a sub-theme under the theme, caregiver burdens.

2. The first quote on page 24 … “when it gets to some point, I think that when parents enter a distressing situation … could have other interpretations. For instance, it may reflect caregivers viewing their role as a form of modelling for their own children. It may also reflect a belief that failure to provide care represents a form of moral transgression, which could be punished indirectly by their own children by neglecting them in the future. For the second quote on the same page, some elaboration would be helpful. For instance, what societal norms obligate the eldest daughter to be a default caregiver?

3. The first quote on page 24 could be interpreted in different ways. It would be helpful if the authors could interpret the "tension" between the caregiver and the husband and what might be driving that tension, as expressed in the quote.

4. On page 26, the statement "Elizabeth shared the difficulty she encounters when she compared toileting assistance with little children and her mother" is not clear. In comparing caring for older adults to children, is this caregiver suggesting that caring for older adults is more difficult than caring for children, or are they the same/similar? The quote supporting the claim also needs clarification. Some additional commentary could help to throw some light on that quote.

Minor issues:

There are a few statements that need clarification. For example, “More recent studies on the motivating factors of adult children to provide support these findings as also identifying reciprocity, sense of obligation, selflessness, and feelings of closeness, secure attachment, and entitlement of care needs as some motivating factors to care for older parents.” (Page 3 paragraph 2)

“Pampers” is a brand of diapers. However, in Ghana, people tend to refer to all kinds of diapers as “pampers”. I suggest the authors indicate the ‘local meaning’ of pampers in the quotation on page 17.

On page 23, the authors need to take another look at this statement “Reciprocity was present at every expression of caregivers’ even up to the point of were expressing their desire to quit caregiving” to ensure clarity.

6. PLOS authors have the option to publish the peer review history of their article (what does this mean?). If published, this will include your full peer review and any attached files.

Reviewer #1: No

Reviewer #2: No

---

## [Author Response · Author response to Decision Letter 0]

28 Feb 2022

Responding to Reviewer’s comments

Thanks for your in-depth feedbacks. 

Reviewer #1: 

This manuscript describes the findings of a phenomenological study of the experiences of caregivers of older adults in Ghana. The authors present a well-written summary of their findings within the context of the Ethics of Care. additional information about the conditions experienced by the care recipients would help to put the caregiving narratives into context given that caregivers of individuals with different chronic conditions experience caregiving and its strain differently. For example, caregivers of people with dementia often rate their strain/burden and stressors higher than those caring for adults with other conditions.

Response: Thank you for being given a high recommendation to manuscript well crafted presentation. Thanks for these important comments. Authors have included the conditions of the care recipients in Table 1.

Reviewer #2: This manuscript presents a qualitative study that explored the lived experiences of caregivers of older adults with functional difficulties. The findings are insightful and the manuscript is also well written. My main concern with the manuscript is the presentation of findings. The findings generally depict detailed experiences of caregivers; however, given the use of the IPA analytical approach, I thought the authors could enrich the presentation by providing a more in-depth interpretation of the interview extracts used to support the narratives. Rather than leaving extracts hanging, it would be helpful if efforts were made in interpreting the subjective meanings of the extracts to the participants.

Response: Thanks for your suggestions. While we appreciate your suggestions, we employed IPA, which give much focus on its iterative nature. Authors believe we have carefully interpreted the findings succinctly and in a manner that conveys their experiences. 

Below are a few (but not the only) instances I thought more could be done with the extracts:

1. On page 18 (last line) it is indicated that “Commitment to caregiving was reflected in the caregivers’ management strategies to combat their own stress …” However, the specific strategies the caregivers employed in managing their stress are not indicated nor elaborated. Moreover, I was wondering why these stress management strategies could not be captured as a sub-theme under the theme, caregiver burdens.

Response: thanks for the suggestions. Please note that extant discussion has been offered on the coping strategies under the theme “Committing the self to caregiving”. This is because commitment references their willingness to cope and keep on with caregiving duties. Ethics of care theory influenced the analysis’ final themes. 

2. The first quote on page 24 … “when it gets to some point, I think that when parents enter a distressing situation … could have other interpretations. For instance, it may reflect caregivers viewing their role as a form of modelling for their own children. It may also reflect a belief that failure to provide care represents a form of moral transgression, which could be punished indirectly by their own children by neglecting them in the future. 

Response: Thanks for your suggestions. We believe the description you have giving is referring to reciprocity, which the authors have carefully interpreted in the manuscript. Authors stated that “The recognition that one day, caregivers may, in turn, require care from their children held great meaning according to caregivers’ narratives. Margaret explained her belief in reciprocity and how that influenced her desire to leave a model for her children”

For the second quote on the same page, some elaboration would be helpful. For instance, what societal norms obligate the eldest daughter to be a default caregiver?

Response: Please see my comments above on our careful interpretation of the findings. The detail discussion of the social norms is being discussed in the discussion. 

3. The first quote on page 24 could be interpreted in different ways. It would be helpful if the authors could interpret the "tension" between the caregiver and the husband and what might be driving that tension, as expressed in the quote.

Response: Thanks. Authors explained the driving force of the tension. For instance, we said that “Naomi emphasised the tension she feels from her husband alluding to her unfulfilled duty as a wife as causing an impediment to care”. In this instance, it was the perceived sense of unfulfilled duty as a wife.

4. On page 26, the statement "Elizabeth shared the difficulty she encounters when she compared toileting assistance with little children and her mother" is not clear. In comparing caring for older adults to children, is this caregiver suggesting that caring for older adults is more difficult than caring for children, or are they the same/similar? The quote supporting the claim also needs clarification. Some additional commentary could help to throw some light on that quote.

Response: This was a mistake in the sentence construction. Authors have corrected this sentence. It now reads “Elizabeth shared the difficulty she encounters when she compared assisting toileting in young children and that of older adults like her mother”. We have also added extra extracts from Elizabeth to make the quote much clearer. 

Minor issues: There are a few statements that need clarification. For example, “More recent studies on the motivating factors of adult children to provide support these findings as also identifying reciprocity, sense of obligation, selflessness, and feelings of closeness, secure attachment, and entitlement of care needs as some motivating factors to care for older parents.” (Page 3 paragraph 2)

Response: This sentence has been clarified. It now reads as “Supporting more recent evidence, the current study identifies reciprocity, sense of obligation, selflessness, and feelings of closeness, secure attachment, and entitlement of care needs as some motivating factors to care for older parents.”

“Pampers” is a brand of diapers. However, in Ghana, people tend to refer to all kinds of diapers as “pampers”. I suggest the authors indicate the ‘local meaning’ of pampers in the quotation on page 17.

Response: thanks for the suggestions. Authors have addressed them accordingly. The quote now reads “She goes to the toilet on herself. Because of this, I make her wear diapers. Her condition has worsened, she cannot even tell me that she will want to urinate or defecate and so she can defecate or urinate on herself. With that, I will have to wash all the bedsheet using Dettol so that there will not be any scent on her.” 

On page 23, the authors need to take another look at this statement “Reciprocity was present at every expression of caregivers’ even up to the point of were expressing their desire to quit caregiving” to ensure clarity.

Response: There is an error. We have corrected this error. The sentence now reads. Reciprocity was present at every expression of caregivers’ even up to the point where expressing their desire to quit caregiving.

---

## [Decision Letter · Decision Letter 1]

13 Apr 2022

Using Ethics of Care as the theoretical lens to understand lived experiences of caregivers of older adults experiencing functional difficulties.

PONE-D-21-02211R1

Dear Dr. Awuviry-Newton,

We’re pleased to inform you that your manuscript has been judged scientifically suitable for publication and will be formally accepted for publication once it meets all outstanding technical requirements.

Kind regards,

Steph Scott

Academic Editor

PLOS ONE

Additional Editor Comments (optional):

Reviewers' comments:

Reviewer's Responses to Questions

**Comments to the Author**

1. If the authors have adequately addressed your comments raised in a previous round of review and you feel that this manuscript is now acceptable for publication, you may indicate that here to bypass the “Comments to the Author” section, enter your conflict of interest statement in the “Confidential to Editor” section, and submit your "Accept" recommendation.

Reviewer #1: All comments have been addressed

Reviewer #2: All comments have been addressed

2. Is the manuscript technically sound, and do the data support the conclusions?

Reviewer #1: Yes

Reviewer #2: Yes

3. Has the statistical analysis been performed appropriately and rigorously? 

Reviewer #1: N/A

Reviewer #2: N/A

4. Have the authors made all data underlying the findings in their manuscript fully available?

Reviewer #1: Yes

Reviewer #2: No

5. Is the manuscript presented in an intelligible fashion and written in standard English?

Reviewer #1: Yes

Reviewer #2: Yes

6. Review Comments to the Author

Reviewer #1: (No Response)

Reviewer #2: (No Response)

7. PLOS authors have the option to publish the peer review history of their article (what does this mean?). If published, this will include your full peer review and any attached files.

Reviewer #1: No

Reviewer #2: No

---

## [Editor Report · Acceptance letter]

18 Apr 2022

PONE-D-21-02211R1 

Using Ethics of Care as the theoretical lens to understand lived experiences of caregivers of older adults experiencing functional difficulties 

Dear Dr. Awuviry-Newton:

I'm pleased to inform you that your manuscript has been deemed suitable for publication in PLOS ONE. Congratulations! Your manuscript is now with our production department. 

Kind regards, 

on behalf of

Dr. Steph Scott 

Academic Editor

PLOS ONE